# Association between inflammatory cytokines and immune–checkpoint molecule in rheumatoid arthritis

**Haruki Matsumoto[1], Yuya Fujita[1], Tomoyuki Asano[1], Naoki Matsuoka[1],
Jumpei Temmoku[1], Shuzo Sato[1], Makiko Yashiro–Furuya[1], Kohei Yokose[1],
Shuhei Yoshida[1], Eiji Suzuki[2], Toru Yago[1], Hiroshi Watanabe[1], Atsushi Kawakami[3],
Kiyoshi Migita[1]***

**1** Department of Rheumatology, Fukushima Medical University School of Medicine, Fukushima, Japan,
**2** Department of Rheumatology, Ohta Nishinouchi General Hospital Foundation, Fukushima, Japan,
**3** Department of Immunology and Rheumatology, Division of Advanced Preventive Medical Sciences,
Nagasaki University Graduate School of Biomedical Sciences, Nagasaki University, Nagasaki, Japan

* migita@fmu.ac.jp

org/10.1371/journal.pone.0260254

**Data Availability Statement:** All relevant data are
within the manuscript and its Supporting
Information files.

**Funding:** The study was supported by the Japan
Grant-in-Aid for Scientific Research(C 17K09981).

## Abstract

### Background

Anti-citrullinated peptide antibodies (ACPA) and inflammatory cytokines play important
roles in the development of rheumatoid arthritis (RA). T cell immunoglobulin and mucin–
domain containing–3 (TIM–3) is an immune-checkpoint molecule involved in inhibitory sig-
naling. Galectin–9 (Gal–9) mediated ligation of TIM–3 induces the amelioration of autoim-
mune diseases. TIM–3 is expressed in synovial osteoclasts and involved in the rheumatoid
bone destruction. The aim of this study was to investigate the relationships between inflam-
matory cytokines and immune–checkpoint molecules in RA patients.

### Methods

Serum levels of interleukin–6 (IL–6), tumor necrosis factor–α (TNF–α), soluble TIM–3
(sTIM–3) and Gal–9 were determined by ELISA. Patients were stratified into two groups
based on ACPA titers: low-medium ACPA (ACPA <200 U/mL) and high ACPA (ACPA ≥200
U/mL). Serum levels of cytokines or immune-checkpoint molecules were evaluated between
RA patients with low-medium ACPA titers and high ACPA titers.

### Results

Elevated serum levels of inflammatory cytokines were correlated with DAS28–ESR in RA
patients. Although serum levels of sTIM–3 were elevated in RA patients, significant correla-
tions between sTIM–3 and cytokines (IL–6 or TNF–α) were observed exclusively in RA
patients with low-medium ACPA titers (<200 U/mL). Serum levels of IL–6 and TNF–α levels
were significantly correlated with elevated Gal–9 levels regardless of ACPA status. A signifi-
cant correlation between IL–6 and Gal–9 was observed in RA patients without advanced
joint damage. Conversely, a significant correlation between TNF–α and Gal–9 was
observed in RA patients with advanced joint damage.

KM is received this grant. URL:https://www.jsps.
go.jp/english/e-grants/ They did not play any role in
the study design, data collection and analysis,
decision to publish, or preparation of the
manuscript.

**Competing interests:** KM has received research
grants from Chugai, Pfizer, and AbbVie. Rest of the
authors declares that they have no competing
interests. They did not play any role in the study
design, data collection and analysis, decision to
publish, or preparation of the manuscript. This
study did not use any of the funds received from
them.

## Conclusions

Our data indicated that there are positive correlations between circulating inflammatory
cytokines and checkpoint molecules in RA patients and these interactions can be modulated
by ACPA status or joint damage stage.

## Background

Rheumatoid arthritis (RA) is an autoimmune inflammatory disease that results in joint
destruction and disability [1]. RA is characterized by the production of auto–antibodies and
cytokine–mediated synovial or systemic inflammation [2]. Cytokine networks play an impor-
tant role in the pathogenesis of RA [3]. Sustained production of proinflammatory cytokines
was shown to correlate with rheumatoid inflammation [4]. Previous studies have shown that
the manipulation of serum cytokine levels is the main mechanism of action of biologics or tar-
geted synthetic disease–modifying antirheumatic drugs (tsDMARDs) for RA [5, 6]. Increased
levels of cytokines, including tumor necrosis factor–α (TNF–α) and interleukin–6 (IL–6),
reflect the rheumatoid synovial inflammation and have been shown to be associated with RA
disease activity or response to anti–cytokine therapy [3, 4, 7, 8]. However, the association
between circulating cytokines levels and RA disease phenotype is not well characterized. TNF–
α plays a key role in the pathogenesis of RA [3]. TNF–α affects different cell types and is pro-
duced mainly by monocytes, macrophages, and T cells [9, 10]. IL-6 is another cytokine which
is considered as disease driving cytokine in RA [11]. In RA, IL–6 is increased in synovium and
blood, and its level correlates with clinical disease activity and joint damage [7, 12].

Anti–citrullinated peptide antibodies (ACPA) are useful for the diagnosis of RA and have
been shown to be associated with progression of joint destruction or therapeutic response in
patients with RA [13]. However, the interactions between ACPA status and proinflammatory
cytokines during RA disease progression are yet to be fully clarified [14]. Rheumatoid synovitis
and articular damage are mediated through several cell types, including T cells, B cells, mono-
cytes, and osteoclasts [15]. Immune–checkpoint molecules may play a role in the interaction
between these immune cells and their activation [16, 17]. T cell immunoglobulin and mucin–
domain containing–3 (TIM–3) is an immune–checkpoint molecule involved in negative regu-
lation of immune responses [18]. Galectin–9 (Gal–9), a ligand of TIM–3, induces the ameliora-
tion of autoimmune diseases [19]. TIM–3 was detected in osteoclasts and its mononuclear
precursor cells in rheumatoid synovium, and the Gal–9/TIM–3 pathway regulatory system
controls osteoclastogenesis and inflammatory bone destruction in RA [20]. In our previous
report, serum Gal–9 and sTIM–3 were significantly elevated in RA patients compared with
healthy subjects [21, 22]. A relationship between proinflammatory cytokines and Gal–9/TIM–
3 has been reported in RA patients [23, 24], so we hypothesized that immune–checkpoint mol-
ecules are possible modulators of immunopathology in RA through their interactions with
cytokines. In this study we investigated the influence of these combined biomarkers, including
autoantibodies, cytokines, and immune-checkpoint molecules, on the disease activity of RA.

## Materials and methods

### RA patients and healthy subjects

This observational single–center study included 132 consecutive RA patients. Patients were
enrolled between February 2012 and June 2020, with follow–up ending in June 2020. We

retrospectively reviewed the records of these RA patients. All patients were treated in Department of Rheumatology, Fukushima Medical School from June 2009 to June 2020. All the patients met the 2010 ACR/EULAR classification criteria for the disease [25]. Probable RA or overlap syndromes; complication of autoimmune/autoinflammatory diseases, cancer, and other diseases that may affect immune checkpoint molecules and cytokines were excluded.

The following clinico–demographic data were collected from the Medical Records Unit at Fukushima University Hospital: age at the time of blood test, age at onset of RA, duration of RA, gender, and Disease Activity Score–28 for Rheumatoid Arthritis with erythrocyte sedimentation rate (DAS28-ESR). As controls, 19 healthy subjects (7 males, 12 females, median age 40 years, interquartile range [IQR]; 33–42 years) were included. This study was conducted in accordance with the principles of the Declaration of Helsinki. Ethical approval for this study (No.2019097) was provided by the Ethics Committee of Fukushima Medical University. In this study, patient's information was collected on a medical record basis and serological bio-markers were measured using stored remaining serum of blood used for daily clinical practice. For this reason, there were many patients who were no longer attending our hospital, and we informed the participants of the study by disclosing the information on our website. This has been approved by the Ethics committee of our hospital.

## Measurement of clinical disease activity

All patients underwent clinical assessment at baseline, including 28–joint swollen and tender joint counts (28–SJC and 28–TJC, respectively), physician and patient global assessment with visual analogue scales (0–100 mm) and ESR (mm/h). The composite disease activity indices were subsequently calculated: DAS28–ESR, the group of $3.2\leq$ DAS28–ESR $\leq5.1$ points was defined as moderate disease activity while the group of $5.1<$ DAS28–ESR was defined as severe disease activity. The anti–CCP antibodies were analyzed using commercially available second–generation chemiluminescent enzyme immunoassay kits (STACIA® MEBLuxTM CCP test, Medical and Biological Laboratories, Aichi, Japan) according to the manufacturer's instructions. The results were reported qualitatively where negative or positive for anti-CCP antibody was defined as $<4.5$U/ml or $\geq4.5$ U/ml, respectively. Radiographs were taken of both hands of each patient.

Two rheumatologists, blinded to the patient's identify and functional status, independently graded each hand radiographs and assigned as Steinbrocker radiographic stage [26]. Stage I is the early RA stage which do not occur joint damage. In Stage II, bone erosion is seen on hand radiographs, but no joint deformity. In Stage III, joint deformity is observed and, in Stage IV, bone ankylosis is observed.

## ELISA methods

Serum concentrations of Gal–9 (No. DGAL90), sTIM-3 (No. DTIM30), IL–6 (No. HS600C) and TNF–α (No. HSTA00E) were measured using enzyme–linked immnunosorbent assay kit (R&D Systems, Minneapolis, MN, USA) according to the manufacturer's instructions. The mean of minimum detectable dose (MDD) in this study was as follows: IL–6, $<0.06$ pg/mL; TNF–α, $<0.092$ pg/mL; Gal–9, $<0.205$ ng/mL; and sTIM–3, $<0.265$ pg/mL.

## Statistical analysis

Results were non–normally distributed and are presented throughout the manuscript with median and interquartile range [IQR], and were compared by the Mann-Whitney $U$ test. Correlations between continuous variables were analyzed by the Spearman's rank correlation test.

All data entry and statistical analyses were performed using SPSS Statistics version 22.0 (IBM, Armonk, NY). In all the analyses, a 2–tailed p < 0.05 was considered statistically significant.

## Results

### Characteristics of patients with RA

Table 1 shows the demographic and clinical data of the 132 RA patients (35 males and 97 females) enrolled in this study. The median age at the blood test was 66 (56–73) years. The median course of RA disease was 7 (2–11) years and median DAS28-ESR levels were 3.0 (2.1–3.8). Fifty–eight (43.9%) patients had moderate or severe disease activity. The proportion of ACPA–negative (<4.5 U/mL) patients was 11% (14 of 132). The proportion of patients with elevated ACPA titers (≥200 U/mL) was 33% (43 of 132). The use of biologics was 31.8% (42 of 132) and subdivided into anti–TNF–α antibody group (n = 17), anti–IL–6 receptor antibody group (n = 10), and others (n = 15).

### Serum levels of cytokines

The baselines serum levels of cytokines and immune-checkpoint molecules in healthy controls were as follows. The median [IQR], minimum and maximum baseline–IL–6 values were 0.432 pg/mL [<0.06–1.295 pg/mL], less than 0.06 pg/mL and 3.645 pg/mL respectively. On the other hand, TNF–α was undetectable in healthy subjects. The median [IQR], minimum and maximum baseline–Gal–9 values were 3.843 ng/mL [3.224–4.559 ng/mL], 2.439 ng/mL and 6.277 ng/mL, and baseline–sTIM–3 values were 870.547 pg/mL [643.592–2058.551 pg/mL], 390.551 pg/mL and 2269.525 pg/mL respectively. To assess the cytokine profiles correspondeing to RA phenotype, we measured the serum levels of IL–6 and TNF–α in RA patients. Serum levels of IL–6 was significantly higher in RA patients (median [IQR], 8.37 [1.87–23.84] pg/mL) than healthy controls (median [IQR], 0.43 [0.06–1.30] pg/mL) (S1 Fig). Although serum TNF-α

**Table 1. Baseline characteristics of 132 Japanese patients with RA.**

| Characteristics | Value |
|---|---:|
| Age at blood test (years), median (IQR) | 66 (56–73) |
| Age at onset (years), median (IQR) | 57 (47–65) |
| Female, n (%) | 97 (73.5) |
| Smoker, n (%) | 48 (36.4) |
| RA–ILD, n (%) | 32 (24.2) |
| Duration of RA (year), median (IQR) | 7 (2–11) |
| ESR (mm/h), median (IQR) | 17 (8.25–30) |
| RF (IU/mL), median (IQR) | 43.5 (11–156.3) |
| Anti CCP–Ab (U/mL), median (IQR) | 58.9 (2.45–344) |
| Corticosteroid, n (%) | 61 (46.2) |
| Methotrexate, n (%) | 68 (51.5) |
| Biologics, n (%) | 42 (31.8) |
| Anti–TNF–α antibody | 17 (12.9) |
| Anti–IL–6 receptor antibody | 10 (7.5) |
| Other | 15 (11.4) |
| DAS28–ESR, median (IQR) | 3.0 (2.1–3.8) |
| Steinbrocker stage | I:39, II:43, III:2, IV:14 |

CCP = cyclic citrullinated peptide, DAS28 = Disease Activity Score, ESR = erythrocyte sedimentation rate,

RF = rheumatoid factor, ILD = interstitial lung disease, IQR = interquartile range.

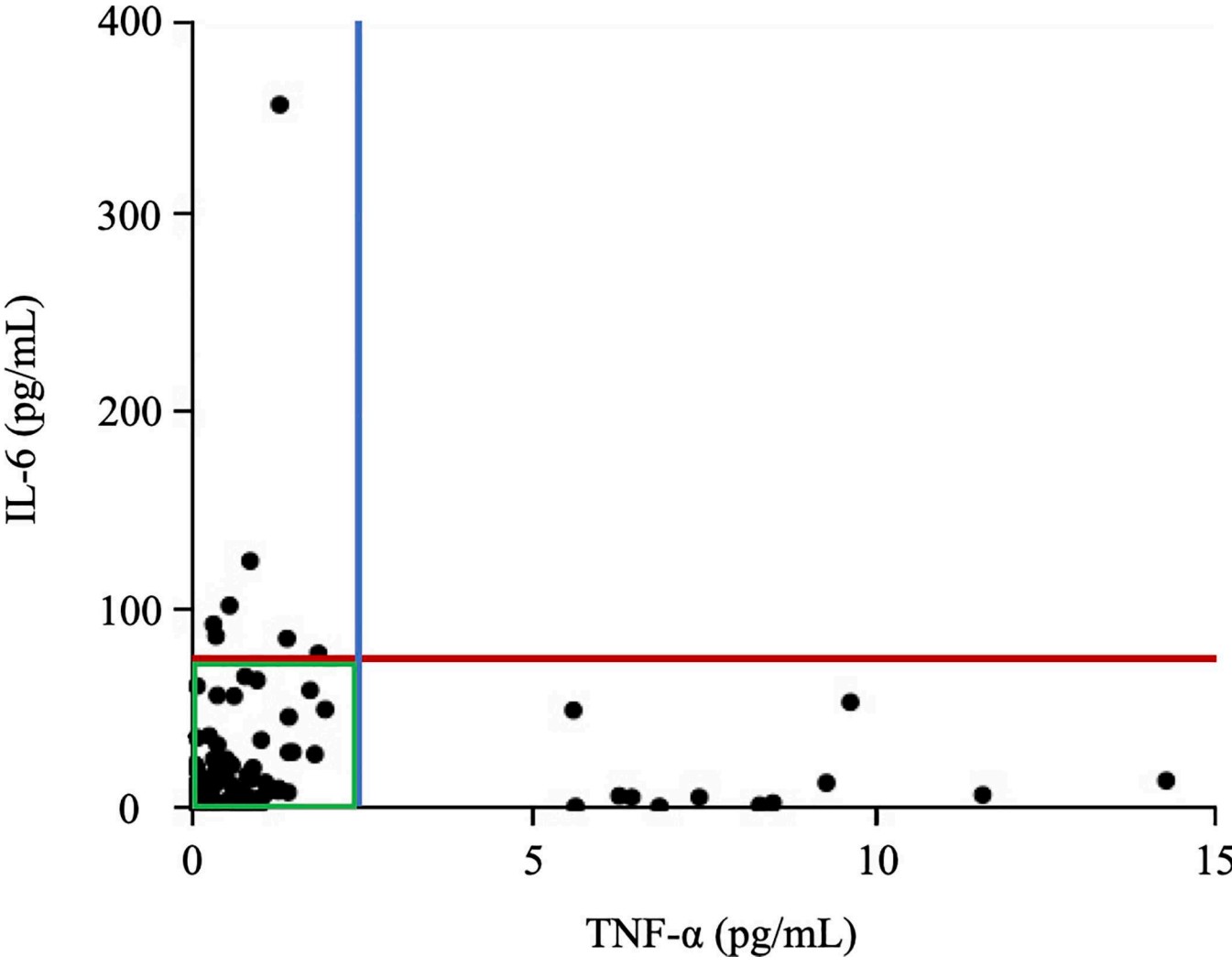

**Fig 1. Relationship between serum levels of TNF–α and IL–6 in RA patients.** The serum levels of IL–6 and TNF–α were elevated and be not correlated with each other.

level was not detected in HC, patients with RA showed elevated serum levels of TNF–α (median [IQR], 0.48 [0.15–0.99] pg/mL). There were no significant differences in serum levels of IL-6 or TNF-a between RA patients with and without RF. Fig 1 shows the relationship between serum levels of IL–6 and TNF–α in RA patients. To confirm the distribution of IL–6 and TNF–α, the patients treated with anti–IL–6 antibodies and anti–TNF–α antibodies were excluded. Interestingly, circulating levels of IL–6 and TNF–α were elevated independently each other; a majority of RA patients presented with high level of IL–6 plus low TNF–α level or high level of TNF–α plus low IL–6 level. Only few RA patients had high levels of both cytokines.

## Serum levels of co-inhibitory checkpoint molecules (S2A and2B Fig)

Consistent with previous reports [21], serum levels of Gal–9 in RA patients (median [IQR], 7.84 [5.69–10.38] pg/mL) were significantly higher than those in HC (median [IQR]; 3.84 [3.22–4.56] pg/mL). Similarly, serum levels of sTIM–3 in RA patients (median [IQR], 2,628

**Table 2. Correlation of baseline serum levels of cytokines biomarkers.**

|  | IL–6 | | TNF–α | | Gal–9 | |
| --- | --- | --- | --- | --- | --- | --- |
|  | rs | P value | rs | P value | rs | P value |
| IL–6 | - | - | 0.275 | 0.001 | 0.326 | <0.001 |
| TNF–α | 0.275 | 0.001 | - | - | 0.358 | <0.001 |
| Gal–9 | 0.326 | <0.001 | 0.358 | <0.001 | - | - |
| sTIM–3 | 0.281 | 0.001 | 0.215 | 0.015 | 0.517 | <0.001 |
| ACPA | -0.044 | 0.620 | 0.029 | 0.744 | 0.294 | 0.001 |
|  | sTIM–3 | | ACPA | | | |
|  | rs | P value | rs | P value | | |
| IL–6 | 0.281 | 0.001 | -0.044 | 0.620 | | |
| TNF–α | 0.215 | 0.015 | 0.029 | 0.744 | | |
| Gal–9 | 0.517 | <0.001 | 0.294 | 0.001 | | |
| sTIM–3 | - | - | 0.236 | 0.007 | | |
| ACPA | 0.236 | 0.007 | - | - | | |

The results were obtained using Spearman's correlation coefficient. ACPA = anti-citrullinated peptide antibodies. Gal–9 = galectin–9, IL–6 = interleukin–6, sTIM–3 = serum T cell immunoglobulin mucin–3, TNF–α = tumor necrosis factor–α.

[1,891–3,664] pg/mL) were significantly higher than those in HC (median [IQR], 871 [644–2,059] pg/mL) [22].

## Correlation of serum levels of cytokines with biomarkers in RA patients

Table 2 shows the correlation of baseline serum levels of cytokines and biomarkers. There was a significant positive correlation between cytokines (IL–6 or TNF–α) and co–inhibitory checkpoint molecules (Gal–9 and sTIM–3). However, there was no significant correlation between serum levels of cytokines (IL–6 or TNF–α) and ACPA titers.

In our previous study, we found that the association between serum levels of Gal–9 or sTIM–3 and ACPA was modulated by the status of ACPA titers [21, 22]. The cutoff value of ACPA titer (200 U/mL) was determined according to the ability to extract the strongest correlation among Gal-9 or sTIM-3 and ACPA titer. We investigated the correlation between circulating cytokines and clinical parameters after dividing RA patients into two groups, based on the presence or absence of high ACPA titers (≥200 U/mL) [21, 22]. As shown in Fig 2, there were significant correlations between serum levels of cytokines (IL–6 or TNF–α) and RA disease activity (DAS28–ESR), and these correlations were not modulated by the high status of ACPA titer (≥200 U/mL). Serum levels of cytokines (IL–6 or TNF–α) were significantly correlated with those of Gal-9 (Fig 3). As shown in Fig 4, there were significant correlations between serum levels of cytokines (IL–6 or TNF–α) and sTIM–3 in RA patients with low–medium levels of ACPA titers (<200 U/mL). However, there was no significant correlation between serum levels of cytokines (IL–6 or TNF–α) and sTIM–3 under the high status of ACPA titers (≥200 IU/mL).

## Relationship between cytokines and Gal–9 according to the joint destruction stage

Serum Gal–9 levels were significantly higher in RA patients with advanced joint damage (stage II–IV) compared to those without joint damage (Stage I) (S3 Fig). We hypothesized that the elevated levels of Gal–9 in RA patients with advanced joint damage were linked with serum

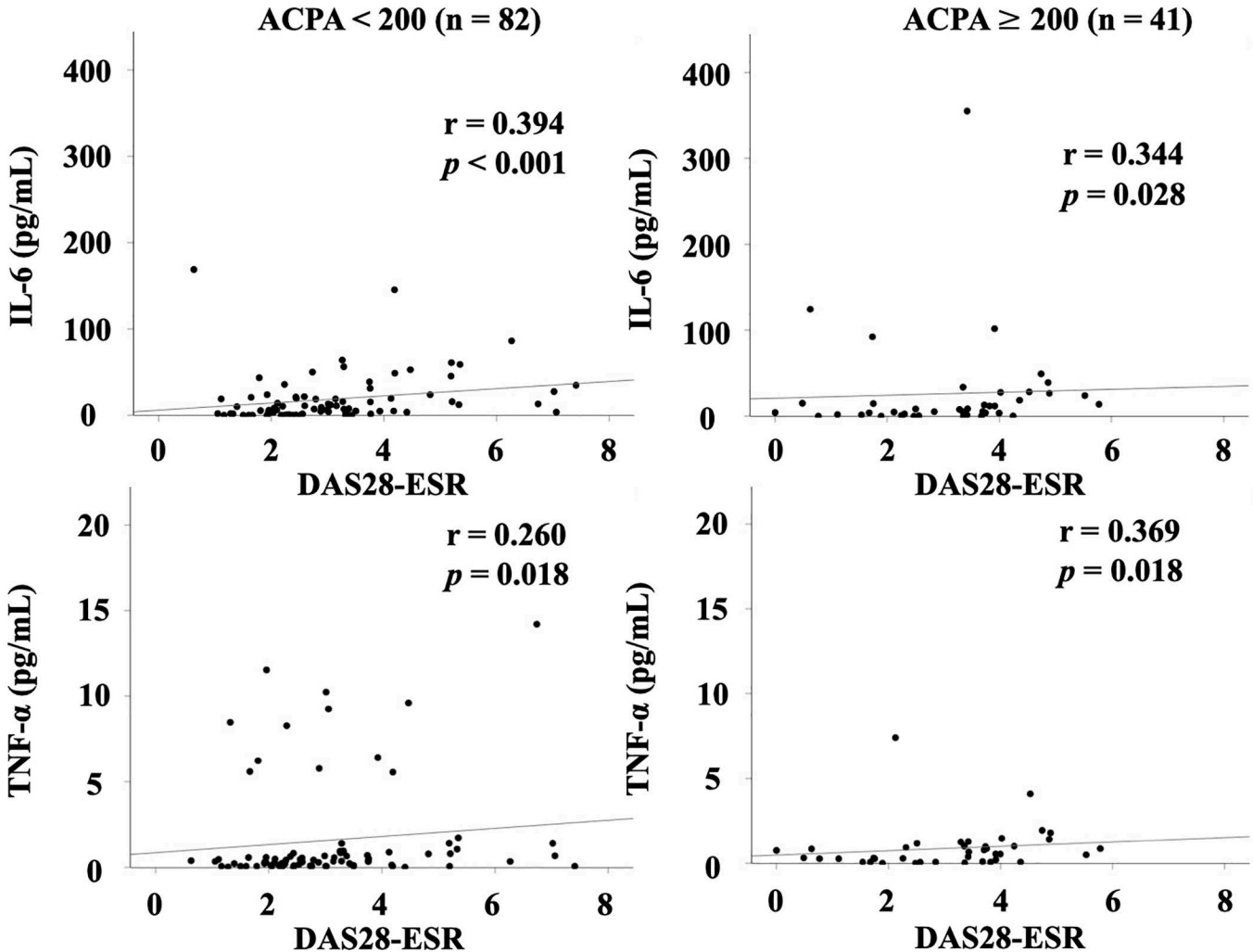

**Fig 2. Relationship between serum levels of cytokines (IL–6 or TNF–α) and RA disease activity (DAS28–ESR) in the sub–grouped RA patients according to the titers of ACPA.** There were significant positive correlations between serum levels of cytokines (IL–6 or TNF–α) and RA disease activity (DAS28–ESR), and these correlations were not modulated by the status of ACPA titer. The correlations were determined using Spearman's rank correlation test.

levels of cytokines. We compared the serum levels of cytokines between RA patients with or without advanced rheumatoid joint damage (Stage I versus Stage II–IV). Serum levels of TNF-α in RA patients with advanced joint damage (Stage II–IV) were significantly higher than those in patients without advanced joint damage (Stage I) (*p* = 0.0265, S4A Fig). Similarly, serum levels of IL–6 in RA patients with advanced joint damage were higher than those in patients without advanced joint damage; however, the difference between the 2 groups was not statistically significant (*p* = 0.1733, S4B Fig).

RA patients with advanced joint damage (Stage II–IV) showed elevated serum levels of Gal–9. Therefore, we assessed the correlation between cytokines and Gal–9 under the differential rheumatoid joint damage stage. As known in Fig 5A, RA patients without advanced joint damage showed a positive correlation between IL–6 and Gal–9. In contrast, there was a significant correlation between TNF–α and Gal–9 in RA patients with advanced rheumatoid joint damage (Stage II–IV) (Fig 5B). This positive correlation between TNF–α and Gal–9 was more markedly observed (Fig 6) in RA patients with high ACPA titers (≥200 IU/mL).

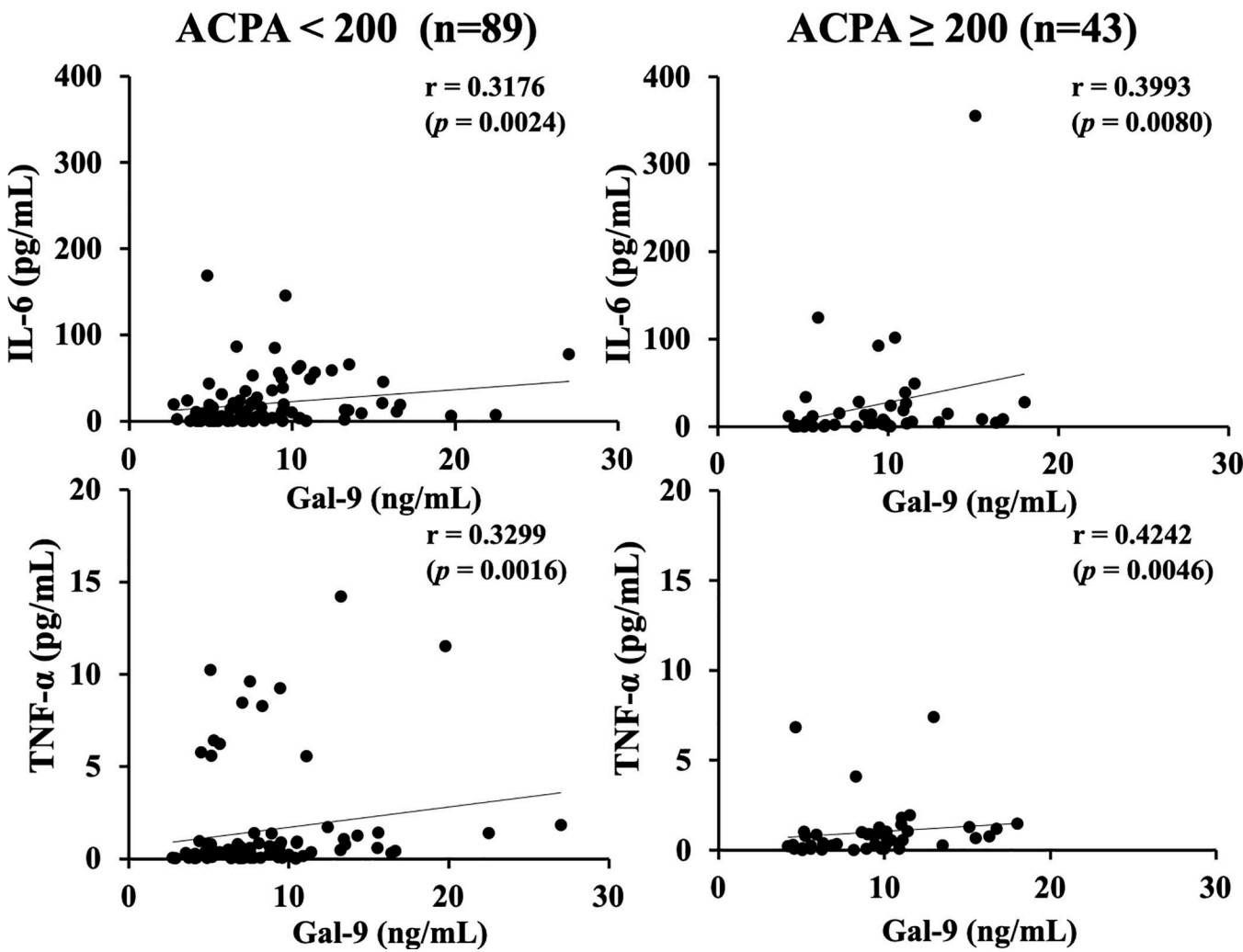

**Fig 3. Relationship between serum levels of cytokines (IL–6 or TNF–α) and Gal–9 in the sub–grouped RA patients according to the titers of ACPA.** There were significant correlations between serum levels of cytokines (IL–6 or TNF–α) and Gal–9, and these correlations were not modulated by the high status of ACPA titers. The correlations were determined using Spearman's rank correlation test.

## Discussion

Dysregulated cytokine production plays an important role in the progression of RA [4]. Given the complexity and heterogeneous nature of RA, it is unlikely that a single cytokine may explain the pathophysiology of RA [27]. Among these cytokines, the interplay between TNF–α and IL–6 is known to play a significant role in the inflammatory processes in RA [3, 4, 7, 8, 28]. Anti–TNF–α therapy resulted in the resolution of elevated levels of IL–6 and acute–phase reactant showing the central role of TNF–α in the pathogenesis of RA [29]. Although IL–6 has not been found to have a crucial role in the effector phase of experimental arthritis, blockade of IL–6 receptor is effective in improving clinical symptoms in RA patients. These findings suggest an independent involvement of IL–6 in rheumatoid inflammation in a fraction of RA patients. Our results indicated the differential regulation of serum levels of TNF–α and IL–6 in patients with established RA. Serum concentrations of TNF–α were relatively low for determination by detection methods contributing to the conflicting outcomes of investigations using RA patients' sera [27]. In this context, we determined the serum TNF–α concentrations using

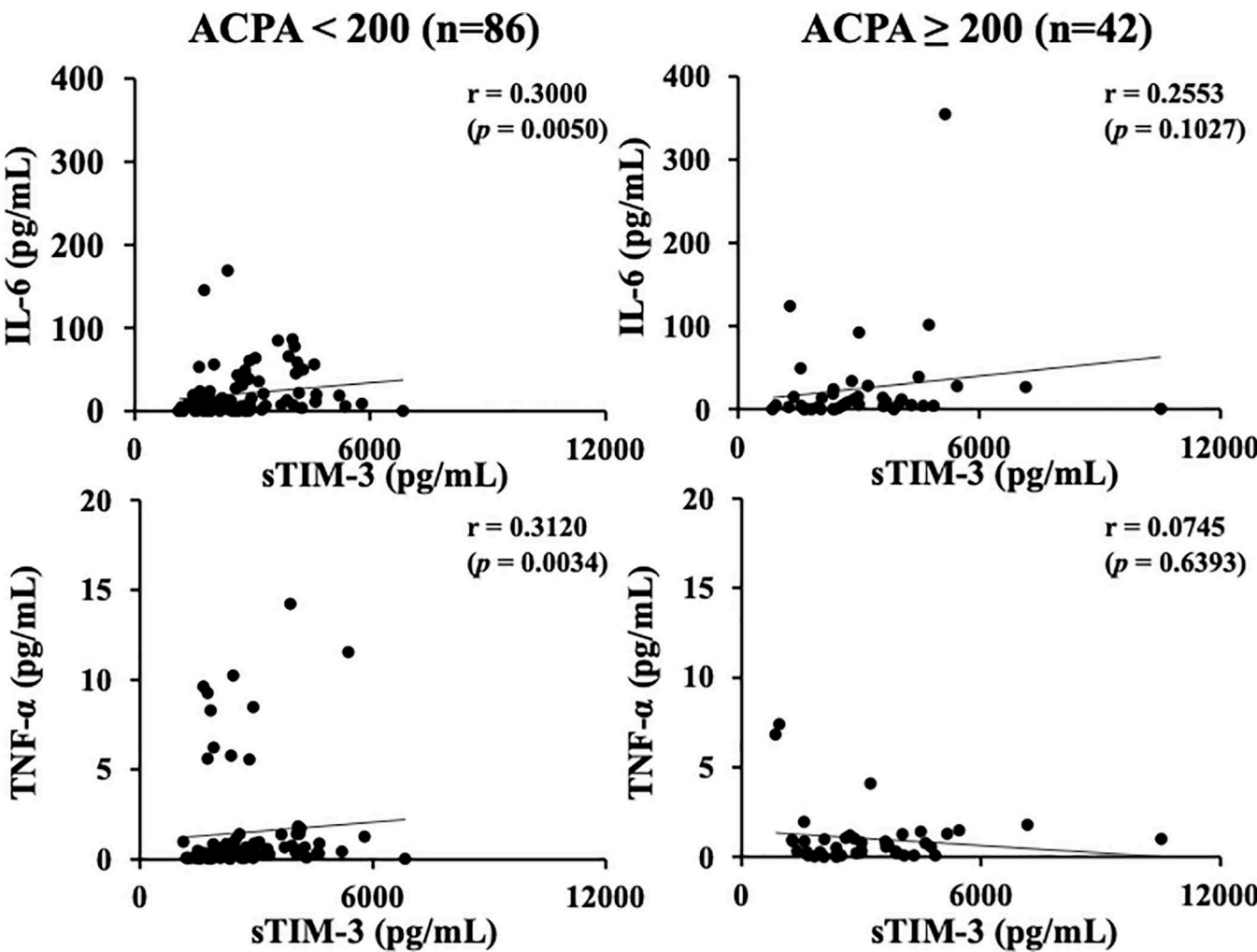

**Fig 4. Relationship between serum levels of cytokines (IL–6 or TNF–α) and sTIM–3 in the sub–grouped RA patients according to the titers of ACPA.**
There were significant correlations between serum levels of cytokines (IL–6 or TNF–α) and sTIM–3 in RA patients with low-medium levels of ACPA titers (<200 U/mL). However, there was no significant correlation between serum levels of cytokines (IL–6 or TNF–α) and sTIM–3 in RA patients with high titers of ACPA (≥200 IU/mL). The correlations were determined using Spearman's rank correlation test.

high–sensitive ELISA method. Our data revealed that majority of RA patients presented with low or marginal levels of serum TNF–α levels, however, circulating TNF–α was detectable in a few patients with RA. In the sub–group of RA patients with low serum levels of TNF–α, IL–6 was randomly distributed and be correlated with rheumatoid inflammatory markers such as ESR. Although there was no significant association between ACPA titers and circulating TNF–α or IL–6 levels, our data demonstrated the associations of these cytokines and co-inhibitory checkpoint molecules under particular ACPA status.

Serum Gal–9 levels were correlated with TNF–α and IL–6, independent of the ACPA titer status. However, elevated levels of sTIM–3 showed positive correlations with these cytokines only in RA patients without high titer of ACPA (<200 U/mL). Shedding of TIM–3 is found LPS–activated CD14$^+$ monocytes or TIM–3 expressing T cell in patients with GVHD after hematopoietic cell transplantation [30, 31]. TIM–3 is a co–inhibitory receptor that is expressed on T cells or innate immune cells, where it has been shown to regulation of their immune responses [32]. TIM–3 expression on CD4$^+$ and CD8$^+$ T cells in peripheral blood or synovial

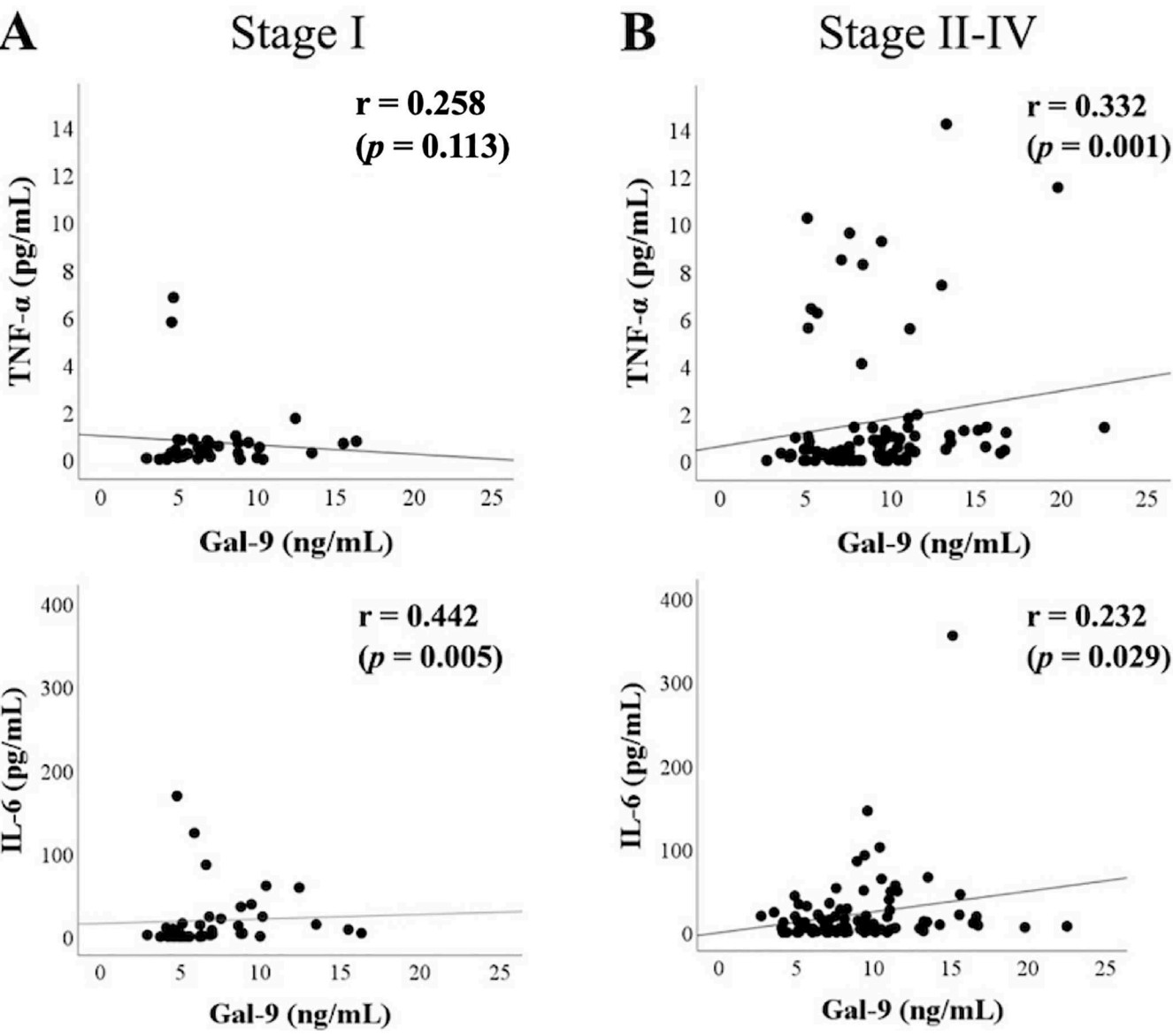

**Fig 5. Relationship between serum levels of cytokines (IL–6 or TNF–α) and Gal–9 in RA patients with or without advanced joint damage.** (A) There was a significant correlation between IL–6 and Gal–9 in RA patients without advanced joint damage (Stage I). (B) Conversely, there was a significant correlation between TNF–α and Gal–9 in RA patients with advanced rheumatoid joint damage (Stage II–IV). The correlations were determined using Spearman's rank correlation test.

fluid was shown to be increased in RA patients [33]. Furthermore, the percentage of TIM–3 expressing CD4$^+$ and CD8$^+$ T cells was negatively correlated with RA disease activity [34]. It was also demonstrated that the expression levels of TIM–3 on T cells was also inversely correlated with plasma TNF levels in RA patients [34]. Taken together, our data suggest that TIM–3 shedding process can be differentially regulated by ACPA status and serum levels of sTIM–3 could be linked with the elevated levels of inflammatory cytokines in RA patients without high titers of ACPA (<200 U/mL).

Additionally, we found that the correlation between Gal–9 and inflammatory cytokines was modulated by the rheumatoid joint damage stage. Strong association of Gal–9 with circulating TNF–α, but not with IL–6, was demonstrated in RA patients with advanced joint damage

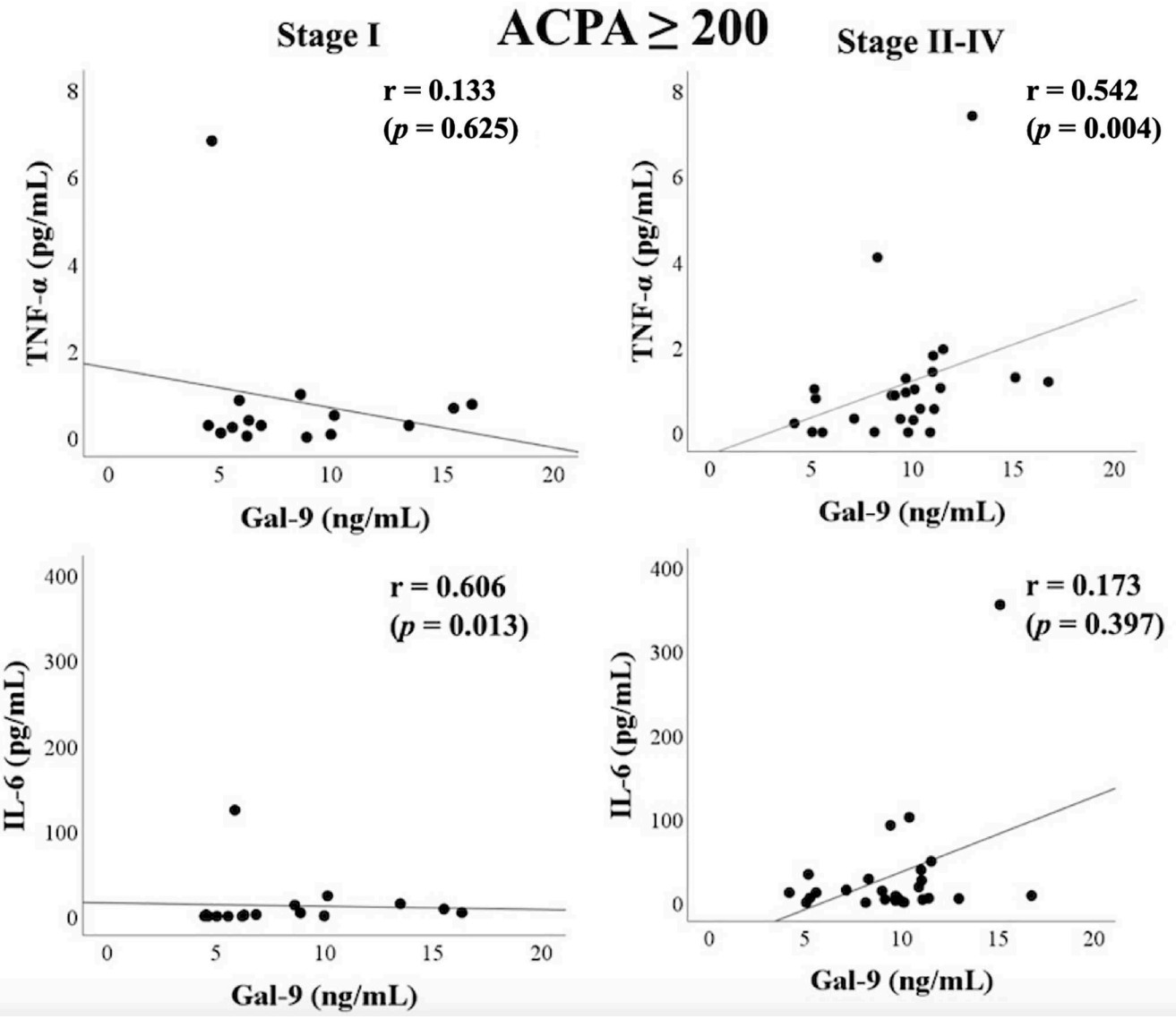

**Fig 6. Relationship between serum levels of cytokines (IL–6 or TNF–α) and Gal–9 in RA patients with high ACPA titers (≥200 IU/mL).** (A) There was a significant correlation between IL–6 and Gal–9 in RA patients without advanced joint damage (Stage I). (B) In contrast, there was a significant correlation between TNF–α and Gal–9 in RA patients with advanced rheumatoid joint damage (Stage II–IV). The correlations were determined using Spearman's rank correlation test.

(Stage II–IV). This tendency was more dominant in RA patients with high titers of ACPA (≥200 IU/mL). Conversely, serum levels of Gal–9 showed a stronger correlation with IL–6 in RA patients without advanced joint damage (Stage I). These findings suggest that the interaction between IL–6 and Gal–9 may be involved in rheumatoid inflammation in the early stage of RA. In contrast, interaction between TNF–α and Gal–9 may be involved in RA progression in advanced stages of joint destruction. The dual role of Gal–9, in correlating with inflammatory cytokines or ACPA titers may define the hierarchical position of RA phenotype or progression. These data suggest that the interplay between inflammatory cytokines and checkpoint molecules may differentially contribute to the rheumatoid inflammatory and articular destruction processes.

Gal–9 and these inflammatory cytokines seem to be upregulated during rheumatoid inflammation [35, 36]. However, the mechanism by which these molecules are regulated under the inflammatory or autoimmune conditions in various RA patients is not understood. Circulating Gal–9 could be a useful biomarker to monitor disease activity and facilitate personalized treatment [37]. Gal–9 is anti–inflammatory as suggested in RA model mice, collagen–induced arthritis, in which Gal–9 injection suppressed osteoclasto genesis through the Gal–9/TIM–3 pathway [20]. Conversely, in RA patients, decreased Gal-9/TIM-3 signaling has been observed since the expression levels of TIM–3 on CD4[+] T cells were lower compared to those from healthy controls [38–40]. Although its function is not fully elucidated, Gal–9 is a potent immune regulator and modulates rheumatoid inflammation and joint destruction [35].

TNF–α was shown to induce the secretion of Gal–9 by mesenchymal stem cells in vitro [41]. Conversely, TNF–α downregulated the surface expressions of TIM–3 on T cells probably by shedding mechanism [28]. Although the expression of TIM–3 on immune cells were not investigated in our study, the co–inhibitory Gal–9/TIM–3 pathways can be downregulated by decoy function of circulating sTIM–3. Gal–9 seems to be reciprocally upregulated to regulate the rheumatoid inflammation; however, sTIM–3, which is elevated in RA patients, can competitive inhibit this co–inhibitory Gal–9/TIM–3 pathway. Further studies are required to elucidate the mechanism by which these cytokines and immune–checkpoint molecules co–modulate the rheumatoid inflammatory processes.

Several potential limitations of this study should be considered while interpreting the results. First, the patient population was relatively small. A larger study is required to provide more definitive evidence. Second, although this study should be designed as age-matched case control study, which could not be performed due to the enrollment of elderly RA patients. Therefore, the possibility of biased matching processes cannot be excluded. Third, all patients with RA and healthy individuals in this study were Japanese; additional studies in other ethnic groups are required to verify these findings. Fourth, a majority of the patients were already undergoing treatment; therefore, our findings may not be generalizable to untreated RA patients. Finally, future studies should examine the longitudinal changes in serum Gal-9, sTIM-3, IL-6, and TNF-α levels in patients with RA and assess their clinical course.

## Conclusions

Serum levels of the co–inhibitory checkpoint molecules were elevated and corelated with circulating inflammatory cytokines, IL–6 or TNF–α in RA patients. These interactions seem to be modulated by ACPA status or RA joint damage stage.

## Supporting information

**S1 Fig. Serum levels of IL-6 among RA patients (n = 132) and healthy controls (HCs, n = 19).** Serum levels of IL-6 in RA patients were significantly higher compared to those in healthy HCs. Statistical significance was determined by Mann-Whitney $U$ test.
(TIFF)

**S2 Fig. Serum levels of Gal-9 and sTIM-3 in RA patients (n = 132) and HCs (n = 19).** (A) Serum levels of Gal-9 in RA patients were significantly higher compared to those in HCs. (B) Serum levels of sTIM-3 in RA patients were significantly higher compared to those in HCs. Statistical significance was determined by Mann-Whitney $U$ test.
(TIFF)

**S3 Fig. Serum levels of Gal-9 between RA patients (n = 132) with or without advanced joint damage (Stage II-IV).** Serum levels of Gal-9 were significantly higher in RA patients

with advanced joint damage (stage II–IV) compared to those without advanced joint damage (Stage I). Statistical significance was determined by Mann-Whitney *U* test.
(TIFF)

**S4 Fig. Serum levels of cytokines (IL-6 or TNF-α) in RA patients with or without advanced joint damage.** (A) Serum levels of TNF-α in RA patients with advanced joint damage were significantly higher than those in RA patients without advanced joint damage. (B) Serum levels of IL-6 in RA patients with advanced joint damage (Stage II-IV) were higher than those in RA patients without advanced joint damage (Stage I); however, there was no significant difference. Statistical significance was determined by Mann-Whitney *U* test.
(TIFF)

**S1 File.**
(DOCX)

## Acknowledgments

We are grateful to Ms Sachiyo Kanno for her technical assistance in this study.

## Author Contributions

**Conceptualization:** Haruki Matsumoto, Yuya Fujita, Atsushi Kawakami, Kiyoshi Migita.

**Data curation:** Haruki Matsumoto, Yuya Fujita, Tomoyuki Asano, Naoki Matsuoka, Jumpei Temmoku, Shuzo Sato, Makiko Yashiro–Furuya, Kohei Yokose, Shuhei Yoshida, Eiji Suzuki, Toru Yago, Hiroshi Watanabe.

**Formal analysis:** Haruki Matsumoto, Yuya Fujita.

**Investigation:** Haruki Matsumoto, Yuya Fujita, Kiyoshi Migita.

**Methodology:** Haruki Matsumoto, Yuya Fujita, Kiyoshi Migita.

**Supervision:** Atsushi Kawakami, Kiyoshi Migita.

**Validation:** Yuya Fujita, Kiyoshi Migita.

**Writing – original draft:** Haruki Matsumoto, Kiyoshi Migita.

**Writing – review & editing:** Haruki Matsumoto, Shuzo Sato, Kiyoshi Migita.

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
