## [Decision Letter · Decision Letter 0]

11 Aug 2021

PONE-D-21-21033

Association between inflammatory cytokines and immune-checkpoint molecule in rheumatoid arthritis

PLOS ONE

Dear Dr. Migita,

Thank you for submitting your manuscript to PLOS ONE. After careful consideration, we feel that it has merit but does not fully meet PLOS ONE’s publication criteria as it currently stands. Therefore, we invite you to submit a revised version of the manuscript that addresses the points raised by the reviewer.

I think that it is of special importance to pay attention to his general recommendation to highlight the main conclusions of your study, peraphs with choosing to delete some graphs.

We look forward to receiving your revised manuscript.

Kind regards,

Pierre Busson, MD, PhD, Res Director

Academic Editor

PLOS ONE

Journal Requirements:

- https://arthritis-research.biomedcentral.com/articles/10.1186/s13075-020-02158-3

- https://journals.lww.com/md-journal/Fulltext/2020/10300/T_cell_immunoglobulin_and_mucin_domain_3_is.44.aspx

In your revision ensure you cite all your sources (including your own works), and quote or rephrase any duplicated text outside the methods section. Further consideration is dependent on these concerns being addressed.

3. Please provide additional details regarding participant consent. In the Methods section, please ensure that you have specified (1) whether consent was informed and (2) what type you obtained (for instance, written or verbal). If your study included minors, state whether you obtained consent from parents or guardians. If the need for consent was waived by the ethics committee, please include this information.

Additional Editor Comments:

To be revised according to the recommendations of the reviewer. Higlight key messages and if necessary reduce the number of graphs.

Reviewers' comments:

Reviewer's Responses to Questions

**Comments to the Author**

1. Is the manuscript technically sound, and do the data support the conclusions?

Reviewer #1: Partly

2. Has the statistical analysis been performed appropriately and rigorously? 

Reviewer #1: Yes

3. Have the authors made all data underlying the findings in their manuscript fully available?

Reviewer #1: Yes

4. Is the manuscript presented in an intelligible fashion and written in standard English?

Reviewer #1: Yes

5. Review Comments to the Author

Reviewer #1: Reviewer Recommendation and Comments for Manuscript Number PONE-D-21-21033

Matsumoto et al. represents measurements of Gal-9, sTim-3, IL-6 and TNFa and radiographic staging in 132 patients with established and actively treated RA.

The key finding is that different associations between serum cytokines and soluble immune-checkpoint molecules is altered by their ACPA-status and in general differ between patients notably on different treatments.

In general, an interesting subject but with the descriptive data included in the present paper it is difficult to draw any robust conclusions on the relationship between serum cytokines, soluble immune-checkpoint molecules and ACPA-status.

A short method sections furthers challenges the conclusions being made.

Interesting data that would be of interest to the field but that in its current form needs revisions to increase the impact and sharpens the conclusions.

MAJOR COMMENTS

• In light of the hypotheses – why measure Gal-9 (which is neither an immune-checkpoint molecule nor a cytokine)? Further Gal-9 is not mentioned in the introduction or background session. Why Tim-3 out of multiple interesting immune checkpoint molecules (PD-1, 4-1BB, OX-40 ect). – please elaborate.

• Due to the cross-sectional study design, it is difficult to relate these measurements to progression of RA as indicated in the last part of the background session. Further, on page 20 in the discussion it is stated that … these data suggests that the interplay between inflammatory cytokines and checkpoint molecules may differentially contribute to the RA inflammation and articular destruction processes and page 3 in the conclusions …They may be usefull markers for predicting the phenotype and personalized treatment of RA.

But since possible interactions, phenotypes or personalized treatment were never studied by the present paper together with the cross-sectional design of the study makes these statements a bit speculative. Association is not the same as causality.

In this light the multiple sections on interactions and possible biomarker potential are a bit speculative and not well substantiated by the data presented.

In general, consider aligning the hypotheses and conclusions to better reflect the descriptive design and results being presented.

• Biologics are grouped without mentioning what specific treatment the patients were given at the time of serum acquisition. This is of key importance since many patients receive anti-IL6 or Anti-TNF that would immensely affect the measurements of exactly these to cytokines. Further CRP is not useful in patient on IL-6 treatments. Please clarify the treatments and consider omitting the anti-IL-6 and TNF treated patients from the analyses.

• Considerable difference between the age of the healthy controls and the patients please comment on the potential role of this in the present paper.

• ELISA: What were the cut-off values? Did you test for interference by rheumatoid factor and heterophilic antibodies? – of particular interest since sero-positive and sero-negative patients are being compared.

(Kragstrup, T. W., Vorup-Jensen, T., Deleuran, B., & Hvid, M. (2013). A simple set of validation steps identifies and removes false results in a sandwich enzyme- linked immunosorbent assay caused by anti- animal IgG antibodies in plasma from arthritis patients. SpringerPlus, 2(1), 1–1. http://doi.org/10.1186/2193-1801-2-263)

• Why make a continuous variable as ACPA-titers into a dichotomic? – please elaborate. Are there any correlation between the titers of ACPA and the cytokines / soluble immune checkpoint molecules or Gal-9? Would be interesting to address?

• TNFα is generally being considered a driver of IL-6 (Charles, P., Elliott, M. J., Davis, D., Potter, A., Kalden, J. R., Antoni, C., et al. (1999). Regulation of cytokines, cytokine inhibitors, and acute-phase proteins following anti-TNF-alpha therapy in rheumatoid arthritis. J. Immunol. 163 (3), 1521–1528.). Please elaborate and discuss the finding that the patients either had high IL-6 or TNF but not both.

• If the different serum cytokines, soluble immune-checkpoint molecules and Gal-9 are combined which combination have the highest predictive value towards high disease activity or structural damage. – This could be a need place for the present paper to increase the value/impact of the results presented.

• Consider to focus on one aspect and reducing the numbers of graphs with relatively limited importance for the conclusions to avoid data overload and blurring of the key massages.

MINOR COMMENTS

•

• MMP3 is mentioned as a biomarker of joint damage page 6 and in table 1 but never mentioned in figures or the result session later in the paper – consider include or omit completely.

• How is moderate or severe disease activity defined?, how is overlap syndrome being defined?

• Consider getting the manuscript corrected grammatically to sharpen small errors?

• Please relate the cytokine and Gal-9 levels to other publications on RA patients measuring these proteins.

• Consider putting your figures in a table to increase the comparison between different correlations and the impact of ACPA and remove figures to supplementary.

• Change the scale on the y-aksis of figure 1 to increase the área between 0-100 were the vast majority of the data is situated.

6. PLOS authors have the option to publish the peer review history of their article (what does this mean?). If published, this will include your full peer review and any attached files.

Reviewer #1: No

---

## [Author Response · Author response to Decision Letter 0]

17 Sep 2021

Response to Journal Requirements.

We wish to express our appreciation to your insightful comments on our paper. The comments have helped us significantly improve the paper.

Comment

 and

Response

I appreciate for your critical comment. It has been reworked to meet the style of PLOS ONE`s style.

Comment

- https://arthritis-research.biomedcentral.com/articles/10.1186/s13075-020-02158-3

- https://journals.lww.com/md-journal/Fulltext/2020/10300/T_cell_immunoglobulin_and_mucin_domain_3_is.44.aspx

In your revision ensure you cite all your sources (including your own works), and quote or rephrase any duplicated text outside the methods section. Further consideration is dependent on these concerns being addressed.

Response

I appreciate your pertinent comment. We have quoted or rephrased as necessary outside the methods section.

Comment

3. Please provide additional details regarding participant consent. In the Methods section, please ensure that you have specified (1) whether consent was informed and (2) what type you obtained (for instance, written or verbal). If your study included minors, state whether you obtained consent from parents or guardians. If the need for consent was waived by the ethics committee, please include this information.

Response

Thank you for your comment. In this study, patient information was collected on a medical record basis and serological biomarkers were measured using stored remaining serum of blood used for medical treatment. For this reason, there were many patients who were no longer attending our hospital, and we informed the participants of the study by disclosing the information on our website. This is also mentioned in the article.

Comment

4. We note that you have stated that you will provide repository information for your data at acceptance. Should your manuscript be accepted for publication, we will hold it until you provide the relevant accession numbers or DOIs necessary to access your data. If you wish 

to make changes to your Data Availability statement, please describe 

these changes in your cover letter and we will update your Data Availability statement to reflect the information you provide.

Response

I apologize for the inconvenience. Minimal data used in this study were prepared and sent to you.

Response to Reviewer.

We wish to express our appreciation to your insightful comments on our paper. The comments have helped us significantly improve the paper.

Major comments.

Comment

• In light of the hypotheses – why measure Gal-9 (which is neither an immune-checkpoint molecule nor a cytokine)? Further Gal-9 is not mentioned in the introduction or background session. Why Tim-3 out of multiple interesting immune checkpoint molecules (PD-1, 4-1BB, OX-40 ect). – please elaborate.

Response

I appreciate your significant comment. TIM-3 was detected in osteoclasts and its mononuclear precursor cells in rheumatoid synovium, and the Gal-9/TIM-3 pathway regulatory system controls osteoclastogenesis and inflammatory bone destruction in RA. In fact, in our previous report, serum Gal-9 and sTIM-3 were significantly elevated in RA patients compared with healthy subjects. Therefore, we also focused on the Gal-9/TIM-3 pathway in this study. This information is also added in this article.

Comment

• Due to the cross-sectional study design, it is difficult to relate these measurements to progression of RA as indicated in the last part of the background session. Further, on page 20 in the discussion it is stated that … these data suggests that the interplay between inflammatory cytokines and checkpoint molecules may differentially contribute to the RA inflammation and articular destruction processes and page 3 in the conclusions …They may be useful markers for predicting the phenotype and personalized treatment of RA. But since possible interactions, phenotypes or personalized treatment were never studied by the present paper together with the cross-sectional design of the study makes these statements a bit speculative.

Association is not the same as causality. In this light the multiple sectionson interactions and possible biomarker potential are a bit speculative and not well substantiated by the data presented.

In general, consider aligning the hypotheses and conclusions to better reflect the descriptive design and results being presented.

Response

In the case of unclear causality, the description was changed to suggest the possibility of an association rather than a causality. As you pointed out, the following sentence in the conclusion is nothing but a conjecture and was deleted from the text. “Measurement of serum IL-6 and TNF-α concentrations in combination with checkpoint molecules may help predict the phenotypic alternations of RA.”

Comment

• Biologics are grouped without mentioning what specific treatment the patients were given at the time of serum acquisition. This is of key importance since many patients receive anti-IL6 or Anti-TNF that would immensely affect the measurements of exactly these to cytokines. 

Further CRP is not useful in patient on IL-6 treatments. Please clarify the treatments and consider omitting the anti-IL-6 and TNF treated patients from the analyses.

Response

Thank you for your significant comment. Biologics was subdivided into anti-TNF-α antibody group (n=17), anti-IL-6 antibody group (n=10), and others (n=15). In addition, we remake Figure 2 by excluded the anti-TNF-α antibody group and anti-IL-6 antibody group. There was no significant difference in the distribution of IL-6 and TNF-α between the new figure and the old Figure 2. In addition, we used the DAS28-ESR instead of the DAS28-CRP to evaluate disease activity because, as you pointed out, the inclusion of the IL-6-treated group would make the evaluation of activity by CRP levels inappropriate.

Comment

• Considerable difference between the age of the healthy controls and the patients please comment on the potential role of this in the present paper.

Response

I appreciate for your comment. The healthy controls were volunteers who collected serum samples from the employees of our institution. Therefore, their ages are younger than peak age of RA patients. As you pointed out, the age of the control group is younger than that of RA patients, so, this item has been added to the limitation section.

Comment

• ELISA: What were the cut-off values? Did you test for interference by rheumatoid factor and heterophilic antibodies? – of particular interest since sero-positive and sero-negative patients are being compared. (Kragstrup, T. W., Vorup-Jensen, T., Deleuran, B., & Hvid, M. (2013). 

A simple set of validation steps identifies and removes false results in a sandwich enzyme- linked immunosorbent assay caused by anti- animal IgG antibodies in plasma from arthritis patients. SpringerPlus, 2(1), 1–1. http://doi.org/10.1186/2193-1801-2-263)

(Kragstrup, T. W., Vorup-Jensen, T., Deleuran, B., & Hvid, M. (2013). 

A simple set of validation steps identifies and remove false results in a sandwich enzyme-linked immunosorbent assay caused by anti animal IgG antibodies in plasma from arthritis patients. SpringerPlus, 2(1), 1-1. http://doi.org/10.1186/2193-1801-2-263)

Response

Thank you for your significant comment. The detection limit of each product used in the ELISA and the detection limit in this study have been added to the text.

Gal-9, sTIM-3, IL-6 and TNF-α was compared between the RF positive and negative groups, but no significant difference was observed between two groups. So, we concluded that there was no RF interference in the ELISA.

Comment

• Why make a continuous variable as ACPA-titers into a dichotomic? – please elaborate. Are there any correlation between the titers of ACPA and the cytokines / soluble immune checkpoint molecules or Gal-9? Would be interesting to address?

Response

ACPA titers do not correlate with cytokine (IL-6 and TNF-α) in this study. Previous reports by our department showed that the correlation between ACPA and immune checkpoint molecules (Gal-9 and sTIM-3) was clearly dichotomized by setting ACPA 200 U/mL as a cut off. Therefore, we judged that the cutoff titer of ACPA 200 U/mL is appropriate for clarify the correlation between cytokines and immune checkpoint molecules in this study.

Comment

• TNFΑ is generally being considered a driver of IL-6 (Charles, P., Elliott, M. J., Davis, D., Potter, A., Kalden, J. R., Antoni, C., et al. (1999). Regulation of cytokines, cytokine inhibitors, and acute-phase proteins following anti-TNF-alpha therapyin rheumatoid arthritis. J. Immunol. 163 (3), 1521–1528.). Please elaborate and discuss the finding that the patients either had high IL-6 or TNF but not both.

Response

I appreciate for your pertinent comment. Anti-TNF-α therapy resulted in the resolution of elevated levels of IL-6 and acute-phase reactant showing the central role of TNF-α in the pathogenesis of RA. Although IL-6 has not been found to have a crucial role in the effector phase of experimental arthritis, blockade of IL-6 receptor is effective in improving clinical symptoms in RA patients. These findings suggest an independent involvement of IL-6 in rheumatoid inflammation in a subset of RA patients. However, the distribution of IL-6 and TNF-α showed that IL-6 and TNF-α were visually elevated independently. As mentioned earlier, this trend was also observed when the anti-IL-6 and anti-TNF-α antibody groups were excluded.

Comment

• If the different serum cytokines, soluble immune-checkpoint molecules and Gal-9 are combined which combination have the highest predictive value towards high disease activity or structural damage. – This could be a need place for the present paper to increase the value/impact of the results presented.

Response

Thank you for your comment. The highest correlation with DAS28-ESR, an index of disease activity, were found for IL-6 (r=0.394, p<0.001) in ACPA<200 U/mL and TNF-α (r=0.260, p=0.018) in ACPA≥200 U/mL. Gal-9 is positively correlated with each cytokine at low and high ACPA titer and may be an indicator of disease activity. Thus, in ACPA <200 U/mL, the combination of IL-6 and Gal-9 may be useful in predicting disease activity and structural damage in patients. In contrast, in ACPA≥200 U/mL, the combination of TNF-α and Gal-9 in patients may be useful. 

Comment

• Consider to focus on one aspect and reducing the numbers of graphs with relatively limited importance for the conclusions to avoid data overload and blurring of the key massages.

Response

Thank you for your comment. As you pointed out, we have narrowed down the data and focused the discussion on important items.

Miner comments.

Comment

• MMP3 is mentioned as a biomarker of joint damage page 6 and in table 1 but never mentioned in figures or the result session later in the paper – consider include or omit completely. 

Response

Thank you for your significant comment. MMP-3 was excluded from the list, including Table 1.

Comment

• How is moderate or severe disease activity defined?, how is overlap syndrome being defined?

Response

I appreciate your comment. The group of 3.2 ≤ DAS28-ESR ≤ 5.1 points was defined as moderate disease activity, and the group of 5.1 < DAS28-ESR was defined as severe disease activity. Together, they are described as moderate or severe activity. Overlap syndrome is a complication of autoimmune/autoinflammatory diseases, cancer, and other diseases that may affect immune checkpoints and cytokines. This is added in methods.

Comment

• Consider getting the manuscript corrected grammatically to sharpen small errors?

Response

Thank you for your comment. Some grammatical corrections have been made.

Comment

• Please relate the cytokine and Gal-9 levels to other publications on RA patients measuring these proteins.

Response

Thank you for your pertinent comment. The previous report on proinflammatory cytokines and Gal-9/TIM-3 was added to Background.

Comment

• Consider putting your figures in a table to increase the comparison between different correlations and the impact of ACPA and remove figures to supplementary.

Response

Thank you for your significant comment. Some figures have been moved to supplementary to clarify the issues.

Comment

• Change the scale on the y-aksis of figure 1 to increase the Área between 0-100 were the vast majority of the data is situated.

Response

I appreciate for your comment. Figure 1 has been corrected as you suggested and moved to supplemental.

---

## [Editor Report · Decision Letter 1]

8 Nov 2021

Association between inflammatory cytokines and immune–checkpoint molecule in rheumatoid arthritis

PONE-D-21-21033R1

Dear Dr. Migita,

We’re pleased to inform you that your manuscript has been judged scientifically suitable for publication and will be formally accepted for publication once it meets all outstanding technical requirements.

Please find below some suggestions of minor corrections.

Kind regards,

Pierre Busson, MD, PhD, Res Director

Academic Editor

PLOS ONE

Additional Editor Comments (optional):

Suggestions of corrections:

- Abstract p.2 line 6 : « …osteoclasts and is involved …» instead of « …osteoclasts and be involved …»

- Abstract p.2 line 7 :”…to investigate the relationships between…”instead of ”…to investigate the association between…”

- Introduction p.4 line 17 “…involved in negative regulation of …” instead of “…involved in negative regulator of …”

- Introduction p.5 line 5 “…A relationship between…” instead of “…The relationship between…”

- Materials and Methods p.7 line 6 “…disease activity while the group…” instead of “…disease activity; and the group…”

- Results p.10 line 9 “…corresponding to…” instead of “…corresponded to …”

- Discussion p.17, line 17 :”…in a fraction of RA patients…” instead of “…in a certain of RA patients…”

- Conclusion p.22, line 5 : “…with circulating inflammatory cytokines…” instead of “…with circulating inflammatory arthritis…”
---

## [Editor Report · Acceptance letter]

10 Nov 2021

PONE-D-21-21033R1 

Association between inflammatory cytokines and immune–checkpoint molecule in rheumatoid arthritis 

Dear Dr. Migita:

I'm pleased to inform you that your manuscript has been deemed suitable for publication in PLOS ONE. Congratulations! Your manuscript is now with our production department. 

Kind regards, 

on behalf of

Dr. Pierre Busson 

Academic Editor

PLOS ONE